# Enhancing the land use efficiency of low-land rice (*Oryza sativa L.*)—Grass pea (*Lathyrus sativus L.*) additive series relay intercropping in North-Western Ethiopia: A farmer's indigenous knowledge

**Endeshew Assefa[1], Yayeh Bitew[2]***

**1** Department of Plant Science, Mizan-Tepi University, College of Agriculture and Natural Resource, Mizan-Tepi, Ethiopia, **2** Department of Plant Science, Bahir Dar University, College of Agriculture and Environmental Sciences, Bahir Dar, Ethiopia

* yayehbitew@gmail.com

## Abstract

In Ethiopia, the facts of farmers' indigenous knowledge-based cropping systems have rarely been investigated through research. A field experiment was conducted during 2021/2022 main cropping season at Fogera Plain with the objective of examining the effect of additive series relay intercropping of grass pea with lowland rice on the grain yield of the component crops and the production efficiency of the cropping system. The experiment consisted of a factorial combination of four seed proportions of grass pea (SPGP) (25%, 50%, 75%, and 100% of the recommended seed rate of sole grass pea) relay intercropped with full seed rates of rice in four rice: grass pea spatial arrangements (SA) (1:1, 2:1, 3:1, and their mixed relay intercropping system). The treatments were arranged in a randomised complete block design with three replications. Data on grain yields of the component crops were collected and analysed using SAS-JMP-16 software. Results revealed that SPGP and SA had no significant effect on rice. The highest grain yield of grass pea was obtained when 25% SPGP was relay intercropped with rice in 1:3 SA (5.10 t ha$^{-1}$). Maximum production efficiency in terms of total land output yield (9.89 t ha$^{-1}$) and land use efficiency (ATER = 1.33), net benefit (33, 5176.79 Birr ha$^{-1}$), marginal rate of return (21,428%), and positive monetary advantage index with lower competitive ratio was obtained when 50% SPGP was relay intercropped with rice in 1:3 SA. Thus, this mixture seems to contribute to the development of sustainable crop production with a limited use of external inputs. Rice intercropping with other staple legume crops under residual soil moisture needs to be tested across locations and years to intensify the production efficiency and profitability of the cropping system.

## Introduction

Nowadays, self-sustaining, diversified, low-input, and energy-efficient agricultural systems like intercropping have been considered the most efficient way to achieve sustainability in

**Data Availability Statement:** All relevant data are within the paper and its Supporting Information files.

**Funding:** The authors received no specific funding for this work.

**Competing interests:** The authors have declared that no competing interests exist.

agriculture by many farmers, researchers, and policymakers worldwide [1–3]. A majority of the world's poor farmers, particularly those located in tropical regions including Ethiopia, still depend for their food and income on multispecies agricultural systems, i.e., the cultivation of a variety of crops on a single piece of land [3, 4]. Among these agronomic concepts, intercropping systems (crop diversity in space and time) involving cereal-legume integrations are the most common traditional cropping systems practised worldwide and can be considered a practical application of ecological principles including biodiversity conservation, plant interactions, and other natural regulation mechanisms [5, 6]. This cropping system is assumed to have potential advantages in productivity, stability of outputs, resilience to disruption, and ecological sustainability [6]. It has also resulted in increased farm production and profitability per unit of land area [7, 8].

Research showed that cereal-legume intercropping is widespread among smallholder farmers in sub-Saharan Africa due to over-yielding resulting from (1) the ability of the legume to cope with soil erosion, (2) increasing water use efficiency [9–11], (3) improving light interception [12], and (4) improving nutrient use efficiency [10, 11], (5) controlling weeds, insect pests, and diseases [13], (6) lodging resistance to prone crops, (7) insurance against crop failure [14], (8) nitrogen transfer in cereal-legume intercropping systems [9], and (9) residual effects of no legume-legume cropping system [15, 16]. Among the various cereal-legume intercropping systems/methods, relay intercropping of grass pea (as a supplementary crop) with low land rice (as a main crop) is one of the most common cropping systems practised by subsistence farmers in Northwestern Ethiopia [17].

Low-land rice and grass pea are among the cereal and legume crops, respectively, grown in Fogera Plain, Northwestern Ethiopia [17]. Currently, rice is considered the millennium crop [18], while grass pea has been a neglected crop for many years in Ethiopia [19]. However, both crops are food security crops, mainly for subsistence farmers [17]. In descending order, *Amhara, Gambela, Southern Nations Nationalities and Peoples Region (SNNPR), Benshangul-Gumz, Oromia, and Tigray* are the major rice producing regions in Ethiopia [20]. The study area (Amhara region) accounts for 32% of the area coverage and 28.10% of the annual production in the country [21]. In Ethiopia, grass pea is widely grown in the northwestern (58%), central (16%), and northeastern (13%) parts of the country. The northern and southeastern parts of the country account for the remaining 13% of grass pea area [22].

Rice as a main crop has been grown in both sole and intercropped with grass pea, while grass pea as a supplementary crop is grown in relay intercropped with rice in the Fogera Plain of northwestern Ethiopia [17]. Relay intercropping is a method of multiple cropping where one crop is seeded into a standing second crop well before harvesting [14]. Some of the benefits of relay intercropping are: (1) optimizes system productivity and land use efficiency through growing two or more crops on the same land in the same cropping season; (2) improves soil quality, which in turn increases the component crop yield due to the presence of N fixation in the system [3, 23].

Relay intercropping of grass peas with low-land rice is an indigenous knowledge practised by local farmers in rice-producing areas of Northwestern Ethiopia. The most important reason that farmers use this kind of intercropping is to get additional yield from the supplementary crop (grass pea) and to improve the soil fertility of their cultivated land for the subsequent cropping season, which in turn increases the productivity of low-land rice [17]. Despite this benefit, this cropping system has been practised only in the imagination of local farmers, and its suitability and profitability under research have not yet been investigated. Farmers practised this cropping system in a mixed additive series relay intercropping system (100% rice seed rate with 50% grass pea seed rate) without any row arrangement. The selection of the proper sowing method, spatial arrangement, and ratio with the application of various competition indices

is crucial for adopting the right intercropping system in this environment. Proper row spacing and row arrangement with the appropriate seed proportion are one of the strategy to increase growth resource use efficacy by reducing the competitions [10, 12]. Therefore, the objective of this study was to determine the appropriate sowing methods, spatial arrangement, and optimum seed proportion of grass pea in additive series relay intercropping of grass pea with low land rice for the highest land use efficiency and profitability with a low competitive ratio.

## Material and methods

### Description of the study area

The field experiment was conducted during 2021/2022 main cropping season at Fogera National Rice Research and Training Centre, Northwestern Ethiopia. The experimental site is located at a latitude of 11° 49′ 55″ and a longitude of 37° 37′ 40″, with an altitude of 1774 to 2516 m above sea level. The historical mean annual rainfall varies from 1216 mm to 1336 mm, with an average minimum and maximum annual temperature of 120°C and 280°C, respectively [18, 17]. Generally, the rainfall in the study area follows a dominantly unimodal distribution, with the main peak in June to September, during which more than 80% of the annual rainfall is received. This experimental area is generally found in the agro-ecological zone of moist *Wayena Dega* (mid land). The farming system of the study areas is characterised by 100% mixed crop-livestock systems [3]. The total annual rainfall collected during the experimental year (2021/2022) was 1199 mm, with a mean annual maximum and mean annual minimum temperature of 28.55°C and 18.65°C, respectively (Fig 1).

Immediately prior to implementing the experiment, composite soil samples for soil nutrient analysis were taken at five points diagonally at 0–20 cm soil depth for baseline information using a soil auger. The collected soil samples were analysed at the Adet Agricultural Research Centre soil testing laboratory to determine some of the important physio-chemical properties of soil. The soil texture, organic matter (OM), organic carbon (OC), total nitrogen (TN),

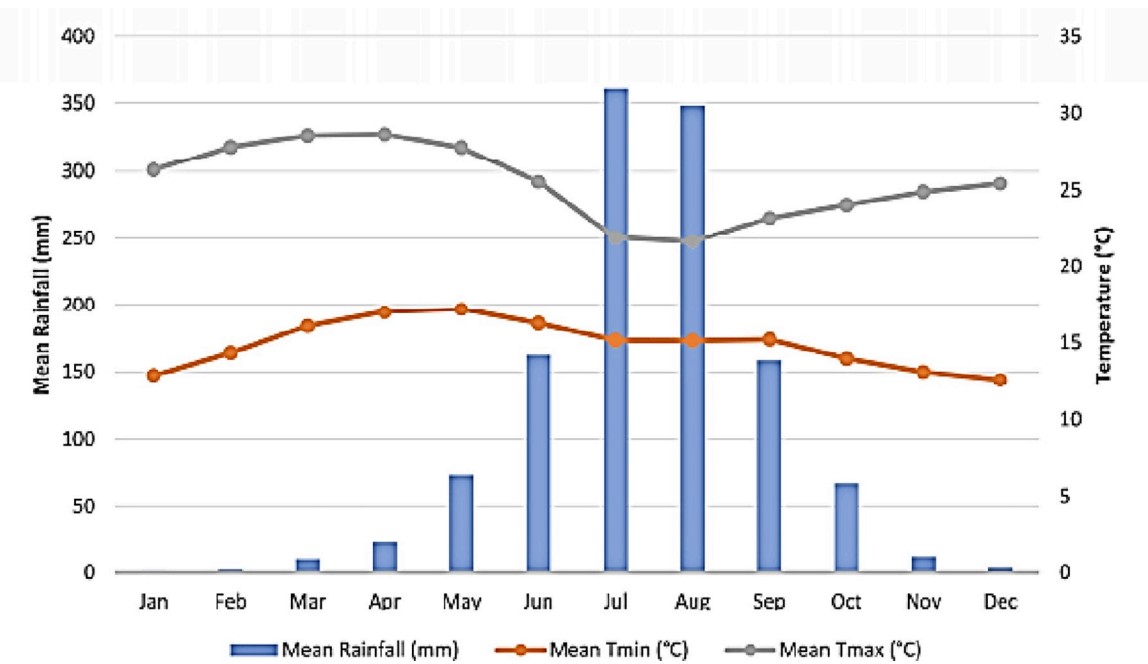

**Fig 1. Monthly mean rainfall, maximum, and minimum temperatures of the experimental site during 2021/2022 cropping season.**

available phosphors (Ava.P), soil reaction (pH), electrical conductivity (EC), and cation exchange capacity (CEC) were determined by Bouyoucos hydrometer method [24], Walkley and Black [25], Walkley and Black [26], Kjeldahl method [27], Bray-II method [28], pH metre [29], Rhoades method [30], and Hesse method [31], respectively. Thus, based on the results of this soil laboratory analysis, the textural class of the soil was found to be clay, with a particle size distribution of 19% sand, 15% silt, and 69% clay. The soil pH value of the experimental site was 5.62, indicating that the soil reactions were found to be moderately acidic [32]. The other soil physio-chemical properties of the study area before planting are described in Table 1 below.

## Experimental test material

The improved cultivar of low land rice *"Shaga"* and a local grass pea variety were used in this experiment as a test crop. The selection criteria for the *"Shaga"* variety are higher yield potential both in research and in the farmer's field and resistance to cold, lodging, and major rice diseases [37]. Local grass pea has wider adaptability and is the default crop selected as it is dominantly grown in association with rice.

## Treatments and experimental design

The treatments consisted of four seed proportions of grass pea (SPGP) (25%, 50%, 75%, and 100% sole grass pea seed rate) relay intercropped with full seed rates of sole rice in (i) three rice (R): grass pea (GP) spatial arrangements (SA) (row ratios) (1:1, 2:1, and 3:1) and (ii) rice (R): grass pea (GP) mixed intercropping (broadcast planting method). Based on the farmers practise, an additive series intercropping experiment design was selected in which rice is the main crop component and grass pea is the supplementary (minor) crop component. Sole rice and sole grass pea were included as comparisons (check) in an experiment. The experiment was laid out in a randomised complete block design (RCBD) with a total of 18 treatments (Table 2) in a factorial arrangement with three replications. The recommended seed rate for sole rice was 100kg ha$^{-1}$ [18]. While the recommended seed rate of sole grass pea was 67kg ha$^{-1}$ [38]. The gross and net plot sizes were 3.4m x 1.5 m (5.1 m$^2$) and 1.8 m x 1.5 m (2.7 m2), with a distance of 0.5 m and 1m between adjacent plots and replications, respectively.

**Table 1. Major soil physical and chemical characteristics of the experimental field before planting.**

| Soil properties | Units | Values | Rating | References |
|---|---|---|---|---|
| **Particle size distribution** | | | | |
| **Sandy** | % | 19 | | |
| **Silt** | % | 15 | | |
| **Clay** | % | 69 | | |
| **Textural class** | | Heavy Clay | | [33] |
| **Soil chemical properties** | | | | |
| **PH** | - | 5.62 | Slightly acidic | [32] |
| **Organic Carbon** | % | 2.2 | Medium | [34] |
| **Organic Matter** | % | 2.61 | Medium | [34] |
| **Total Nitrogen** | % | 0.21 | Medium | [34] |
| **Available Phosphorus** | Ppm | 9.85 | Medium | [35] |
| **Electrical Conductivity** | ds/m | 0.05 | Low | [36] |
| **Cation Exchange Capacity** | C/molg | 57 | High | [32] |

**Table 2. Treatment combination used in additive series relay intercropping of grass pea with rice.**

| Trt No. | SPGP (%) | SA (R: GP) | Type of cropping system |
|---|---|---|---|
| 1 | 25 | 1:1 | Row intercropping |
| 2 | 25 | 2:1 | Row intercropping |
| 3 | 25 | 3:1 | Row intercropping |
| 4 | 25 | MRI | Mixed intercropping |
| 5 | 50 | 1:1 | Row intercropping |
| 6 | 50 | 2:1 | Row intercropping |
| 7 | 50 | 3:1 | Row intercropping |
| 8 | 50 | MRI | Mixed intercropping |
| 9 | 75 | 1:1 | Row intercropping |
| 10 | 75 | 2:1 | Row intercropping |
| 11 | 75 | 3:1 | Row intercropping |
| 12 | 75 | MRI | Mixed intercropping |
| 13 | 100 | 1:1 | Row intercropping |
| 14 | 100 | 2:1 | Row intercropping |
| 15 | 100 | 3:1 | Row intercropping |
| 16 | 100 | MRI | Mixed intercropping |
| 17 | Sole rice | - | Row planting |
| 18 | Sole grass pea | - | Broadcasting |

Trt No., treatment number; SPGP (%), seed proportion of grass pea; SA, spatial arrangement; GP, Grass pea; R, rice; R: GP, rice to grass pea seed proportions; MRI, mixed relay intercropping.

## Experimental procedure and cultural practices

The experiment was conducted under rainfed conditions. Experimental plots were ploughed twice using a tractor. The first plough was done during the dry season in January, and the second plough was done at the end of May. Smoothing and levelling of the experimental plots was done mechanically on the same date that planting was done, June 20, 2021. Subsequently, rice planted in sole and intercropped with grass pea was sown with a seed rate of 100 kg ha$^{-1}$ in 20cm drill row spacing. After 120 days of rice sowing, one row of the associated supplementary crop (grass pea) was sown using the drill row panting method after one, two, and three rows of rice in 1:1, 2:1, and 3:1. All treatments containing rice-grass pea mixed intercropping were also done after 120 days from the rice sowing date. Sole grass pea were planted in the broadcasting planting method. For rice, a fertiliser rate of 46kg N ha$^{-1}$ in the form of UREA and 38 kg P$_2$O$_5$ in the form of NPS were applied [18]. None of the fertilizer types were applied to grass pea in all types of cropping systems. The full dose of P$_2$O$_5$ and 1/3 of N were applied at rice planting, while the remaining 2/3 of N was applied at the flowering stage. All necessary cultural and agronomic practices were carried out uniformly for all plots as per the recommendation for each of the component crops.

## Data collection and measurements

**Grain and straw yield.** The above-ground biomass in the net plot area was manually harvested, collected, and sun dried at 27°C until a constant dry weight was obtained in order to calculate the grain and straw yields of both the component crops. Above-ground biomass that had been dried out was threshed and divided into yields of grain (seed) and straw. After correcting the moisture content of the rice grain and the grass pea seed to 12.5% and 13%,

respectively, grain/seed yields were calculated. The formula shown below was used to calculate the moisture correction factors [39, 40]:

$$\text{Adjusted Yield (t ha}-1) = \frac{100 - \text{Measured moisture content \%}}{100 - \text{MC\%}} \quad (1)$$

Where MC% refers to the moisture content of the component crops. Therefore, adjusted MC% of grain/seed yield = moisture correction factors X grain/seed yield obtained from each plot.

**Competition ratio (CR).**   The term "competition ratio" refers to a measurement of intercrop competition that counts the instances in which one component crop outcompetes the other [41]. Despite the fact that there are numerous indices available to compare interspecific competition in an intercropping system, none of them define the competition effect of the component crops [42] as efficiently as CR. As a result, the competition ratio [43] was used to calculate the degree of competition between the component crops in this cropping system:

$$\text{CRa} = \left(\frac{\text{PLERa}}{\text{PLERb}}\right) x \left(\frac{\text{Zba}}{\text{Zab}}\right) \quad (2)$$

$$\text{CRb} = \left(\frac{\text{PLERb}}{\text{PLERa}}\right) x \left(\frac{\text{Zab}}{\text{Zba}}\right) \quad (3)$$

$$\text{where, PLERa} = \frac{\text{YaIC}}{\text{YaSC}} \text{ and } \text{PLERb} = \frac{\text{YbIC}}{\text{YbSC}} \quad (4)$$

Where, CRa and CRb are competitive ratios of crop *a* and *b*, respectively; PLERa and PLERb are partial land equivalent ratios of crop *a* and *b*, respectively; and Zab and Zba are the seed proportion of crop *a* in an intercropped with crop *b* and the seed proportion of crop *b* in an intercropped with crop *a*, respectively. YaIC and YbIC are the yields of crop *a* and *b* in an intercropping, respectively, and YaSC and YbSC are the yields of crop *a* and *b* in sole cropping, respectively. If CRa is greater than one, it indicates that crop *a* was a competitor, while CRa less than one implies that crop *b* is suppressed crop *a* production.

## Determination of production efficiency

**Land equivalent ratio (LER).**   The effectiveness of land utilisation in an intercropping experiment is gauged by the land equivalent ratio (LER). In comparison to monocropping, it shows how effectively intercropping uses environmental resources [43, 44]. The Mead and Willy [44] formula was used to determine the LER.

$$\text{LER} = \sum_{i=1}^{n} \text{PLERa} + \text{PLERb} + \text{PLERc} \ldots \ldots \ldots \ldots + \text{PLERn} \quad (5)$$

Where, n is the number of crops intercropped in the same areas in one growing season. The value of unity is the critical value. The null hypothesis was LER equal to one, indicating complementarity between two crops. When the LER was greater than one, intercropping favoured the growth and yield of the component crops. In contrast, when LER was lower than one, intercropping negatively affected the growth and yield of the component crops [19, 43].

**Area time equivalent ratio (ATER).**   Area time equivalent ratio provides a more realistic comparison of the yield advantage of intercropping over monocropping in terms of time taken

by component crops in the intercropping systems than LER. The Area Time Equivalent ratio was calculated by a formula developed by Hiebsc [5, 45], as cited by Bitew et al. [3]:

$$ATER = \frac{\sum_{i=1}^{n}(PLERaxTa) + (PLERbxTb)\dots\dots\dots\dots\dots\dots + (PLERnxTn)}{T} \quad (6)$$

Where Ta, duration (in days) of crop *a* to reach maturity; Tb, duration (in days) of crop *b* to reach maturity and *T*, Total duration of the intercropping system (in days) to reach maturity. The interpretation of ATER is the same as with that of LER,

**Rice equivalent yield (REY).** It is the conversion of crop yields into one form that allows us to compare the crops grown under mixed cropping, intercropping, or sequential cropping [46]. The total of intercrop crop *a* yield and converted crop *b* yield was used to compute crop *a* equivalent yield (*a*EY), which was then compared to the sole crop *a* yield. The conversion is done in the form of crop *a* equivalent yield by considering crop *b* yield and the market price of crop *a* and crop *b* [47].

$$aEY \ (t\ ha - 1) = \left(\sum_{i=1}^{n} Yb\left(\frac{Pb}{Pa}\right) + Yc\left(\frac{Pc}{Pa}\right)\dots\dots\dots\dots Yn\left(\frac{Pn}{Pa}\right)\right) + Ya \quad (7)$$

Where, a, b, c. . .. . .n are type of crops grown in an intercropping; Y, yield; P, pric

**Total land output (TLOY).** Total land output yield (TLOY), another measure of intercrop productivity, was used in the evaluation. Total land output yield evaluates total production through a mixture without regard to species combinations or densities [48]. Plots with intercropping that had higher TLOY values than monocultures demonstrated a yield benefit. This is how the TLO was calculated:

$$TLOY \ (t\ ha^{-1} =) \sum_{i\ 1}^{n} Ya + Yb + Yc\dots\dots\dots Yn \quad (8)$$

## Economic analysis

**Monetary advantage index (MAI).** It was estimated to provide a financial comparison between intercropping and single cropping. The Willey formula [49] was used to calculate the monetary advantage index (MAI).

$$MAI = (\sum_{i=1}^{n} a + b + c\dots\dots..n) * \frac{LER - 1}{LER} \quad (9)$$

Where, a, b, c. . ..n are component crops in an intercropping system.

## Partial budget analysis

Following the CIMMYT [50] methodology, a partial budget analysis was carried out to identify the treatment(s) that would be economically profitable. Input costs included the cost of DAP fertiliser and labour for planting grass peas, as well as the cost of harvesting, discarding, and cleaning both component crops. Only major treatments are considered for partial budget analysis, according to CIMMYT [50]. Thus, only grass pea straw and seed yield were considered income elements, as rice straw and grain yields were not significantly affected by the treatments. Other production costs, such as labor for land preparation, weeding of rice, and fertilizer application for rice, were considered fixed costs across treatments. All costs were calculated as the average value of 2021 and 2022 on a per-hectare basis. The price of DAP was 17 birr kg$^{-1}$, and the average labor cost was estimated at 100 Ethiopian birr (ETHB) per man per day. The expenses of grain and straw yield for grass pea were 44 and 2 ETHB kg-1,

respectively, based on the average local market prices for the months of June through February. In the same months, the average cost of grain and straw yield for rice was 60 and 3 ETHB kg-1, respectively. The yields of the treatments were reduced to 90% in order to close the yield gap between experimental plots and farmers because experimental plot yields are typically higher than those of farmers. Using this data, the following formulas were used to compute the total variable cost (TVC), gross benefit (GB), net benefit (NB), and marginal rate of return (MRR):

Total variable cost (TVC):

$$\text{TVC (ETHB)} = \sum_{i=1}^{n} C1 + C2 + C3 \ldots\ldots\ldots\ldots\ldots Cn \tag{10}$$

Gross benefit:

$$\text{GB (ETHB)} = \sum_{i=1}^{n} U1 + U2 + U3 \ldots\ldots\ldots\ldots\ldots Un \tag{11}$$

Net benefit:

$$\text{NB (ETHB)} = GB - TVC \tag{12}$$

After treatments were arranged in ascending order by TVC value, treatments with a high NB and a lower TVC than the preceding treatment were selected for further analysis. Thus, treatments with a lower NB value and a greater TVC than the preceding were excluded. Selected treatments were subjected to marginal rate of return (MRR) analysis, which was calculated by the following formula:

$$\text{MRR (\%)} = \frac{NBT2 - NBT1}{TVCT2 - TVCT1} x100 \tag{13}$$

Where T2 and T1 are consecutive treatments (T) arranged in ascending order based on their TVC after excluding treatments with low NB and high TVC.

## Statistical data analysis

Quantitative information on the experimental field's component crops was entered into Microsoft Office Excel. The data management system was the same. Statistical software was used to assess the component yield data [51]. The data were examined for normal distribution before analysis using the scatter plot method. Grass pea seed percentage and spatial arrangement were fixed variables in the analysis, while replication was a random effect. First, an analysis of variance (ANOVA) using the single degree of freedom orthogonal contrasts approach was performed on the data analysis of the yield of the component crops in all treatments (18). The Tukey-Kramer HSD test was used to separate the means of treatments where there were significant differences between them at any probability level. However, if there was no significant difference (*P > 0.05*) between all treatments (18), the data analysis of the same data in sixteen treatments (excluding the sole crops) was subjected to analysis of variance (ANOVA) following the same procedure. Mean separation was done using the same test when there were significant differences between sixteen treatments at any probability level. A regression analysis was carried out to examine the relationship between production efficiency (total land output and rice equivalent yield) and seed proportion and spatial arrangement.

## Result and discussion

### Grain (seed) yield of the component crops

Analysis of variance showed that the main effects of seed proportions of grass pea (SPGP) and spatial arrangements (SA) and their interaction effects did not significantly ($P > 0.05$) affect rice grain production (Fig 2). In this experiment, the rice grain yield ranged from 4.25 t ha-1 when grass pea seed rates of 100% and 50% were intercropped with rice in 1:3 SA and 1:2 SA, respectively, to 5.32 t ha-1. This result is in conformity with the results of Yayeh and Fikremariam [17], who reported that all the agronomic attributes of rice were not significantly *(P > 0.05)* affected by the main effects of seed proportions of chickpea and spatial arrangements and by their interaction effects. This might be due to the fact that early planting of rice in a relay intercropping system takes advantage of peak resource demands for nutrients, water, and sunlight for all treatments. The competition ratio (CR) of rice in grass pea-rice intercropping, as indicated in Fig 3, confirms these remarks. Moreover, Bitew et al. [3] and Isaac et al. [52] also revealed that all the yield and yield components of maize were not significantly *(P > 0.05)* affected by the main effects of seed proportion of haricot bean and spatial arrangement and by their interaction.

On the other hand, grain yield of grass pea was significantly *(P < 0.05)* affected by the main effect of seed proportion of grass pea and spatial arrangements and by their interaction (Fig 2). Maximum grain yield (5.06 t ha$^{-1}$) was obtained when 25% SPGP was relay intercropped with rice in 1:3 SA. This cropping system gave a 7% higher grass pea seed yield over the sole cropping system. However, it has a statistically similar effect with the treatments treated with the same SPGP with 1:2 SA (4.94 t ha$^{-1}$), mixed relay intercropping (MI) (4.89 t ha$^{-1}$), 50% (4.90 t ha-1), and 100% (4.99 t ha$^{-1}$), SPGP was relay intercropped with rice in 1:3 SA. While,

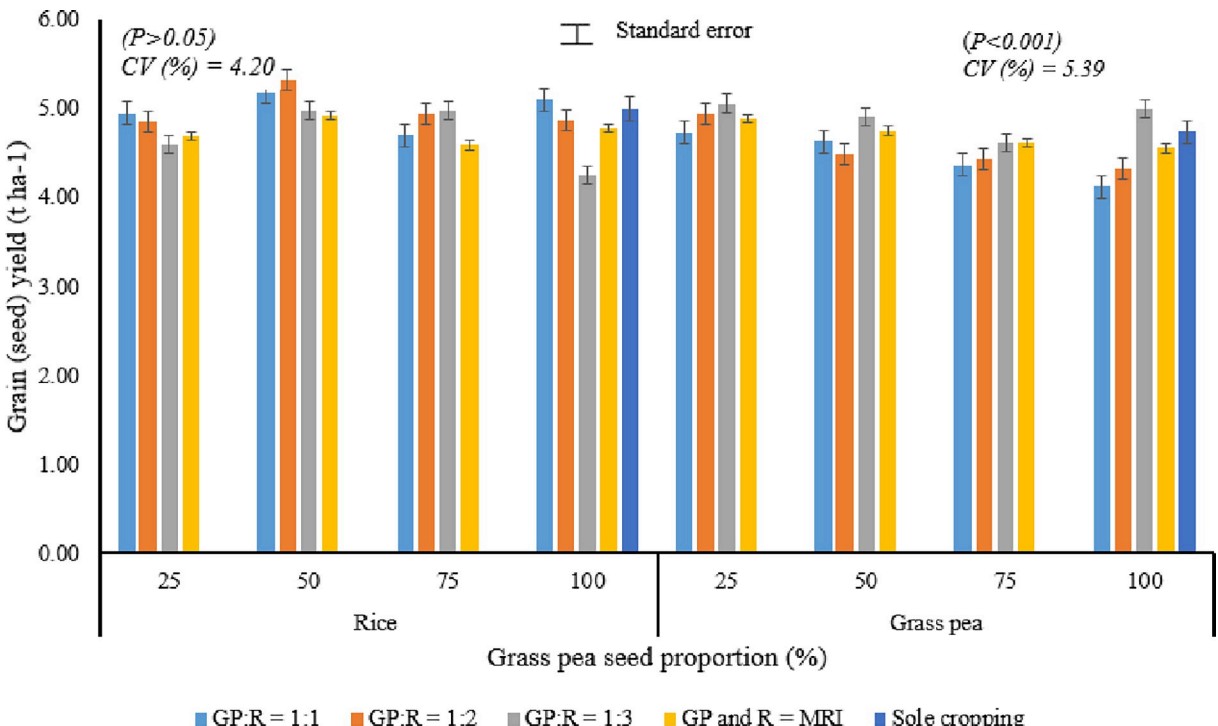

**Fig 2. Interaction effects of SPGP (%) and (SA) in additive series relay intercropping of grass pea (GP) with rice (R) on the component crops economic yield.** GP: R, 1:1, 1:2, and 1:3, indicating that grass pea was planted after one, two, and three rows of rice, respectively; GP and R, MRI, indicating that grass pea was mixed relay intercropped with rice.

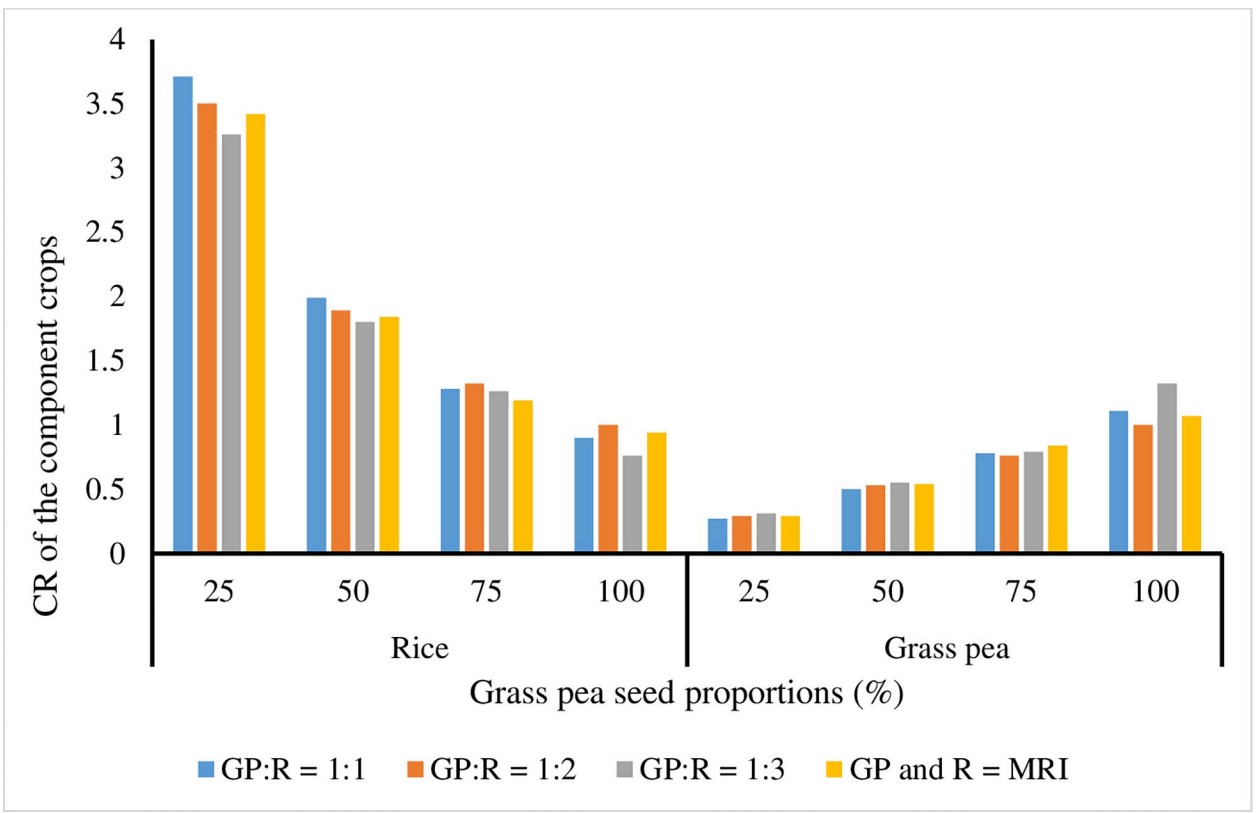

**Fig 3. Competitive ratio of the component crops in grass pea-rice relay intercropping.**

minimum grain yield was obtained when 100% SPGP was relay intercropped with rice in 1:1 SA (4.12 t ha$^{-1}$) (Fig 2), The higher grain yield in the former cropping system might be due to the lower plant population that resulted from the wider plant spacing between rows of grass pea that caused lower inter- and intra-plant competition. This result contradicted the results of Banik et al. [53], who observed that the chickpea yield in a wheat-chickpea mixture was significantly lower than that of a solely cropped chickpea.

**Competition ratio (CR).** Results showed that in all SA, rice had a CR greater than one, with the exception of areas where grass pea was intercropped with rice at a 100% seed rate (Fig 3). This can be because rice and grass pea co-grew with very little resource use from grass pea (mostly). Therefore, rice is the crop that is more competitive in terms of growth resource usage, especially light, when compared to grass pea (rice caused a shadowing impact that in turn influenced the growth and production of grass pea) (Fig 3). The CR of rice was higher than the CR of grass pea. This outcome was consistent with the findings of Yayeh et al. [54], who found that in the local lupine-cereal intercropping system, wheat had a greater competitive ratio than local lupine. Similar to this, when finger millet and haricot bean were intercropped, their vast temporal niche differentiation lessened competition [19].

**Land use efficiency.** The findings showed that rice and grass pea had partial land equivalent ratios (PLER) greater than 0.5 in all cropping systems (Fig 4). This supported the CR values since it showed that there was no or very little resource rivalry amongst the component crops during their co-growing time. This result conflicts with that of Chen et al. [55], who found that the PLER for sorghum and cowpea in a sorghum-cowpea intercropping system was lower and greater than 0.5, respectively, indicating a disadvantage for cowpea and an advantage for sorghum in an intercropping.

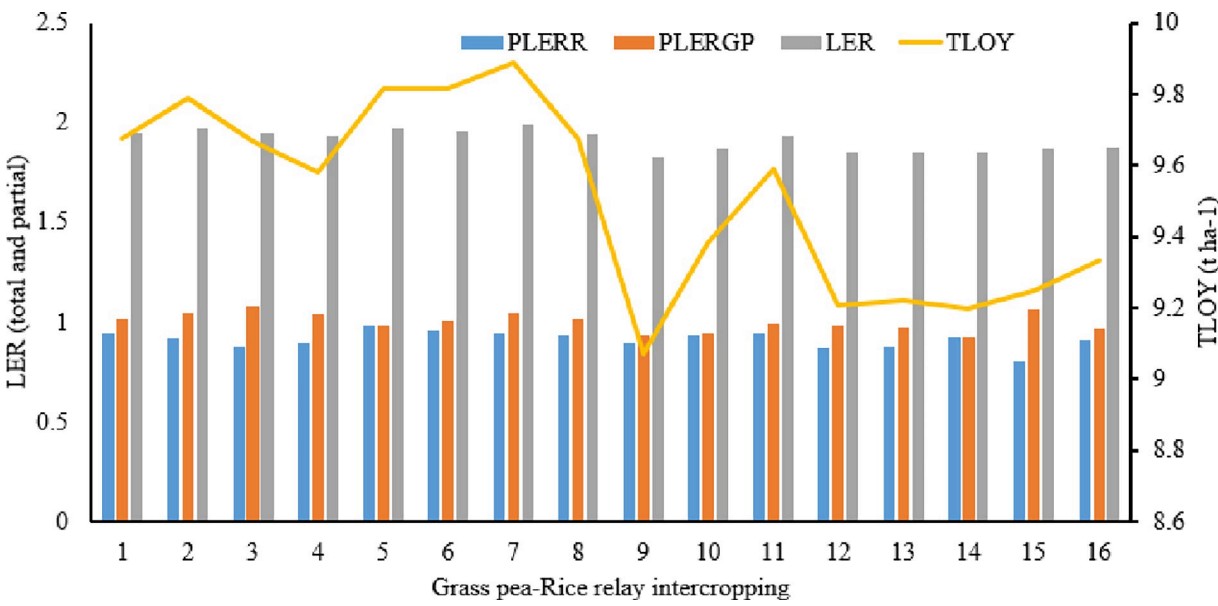

**Fig 4. Land equivalent ratio (LER) (total and partial) and total land output yield (TLOY) in additive series relay intercropping of grass pea with rice.** Note: 1 = 25%+1:1, 2 = 25%+2:1, 3 = 25%+3:1, 4 = 25%+M, 5 = 50%+1:1, 6 = 50%+2:1, 7 = 50%+3:1, 8 = 50%+M, 9 = 75%+1:1, 10 = 75%+2:1, 11 = 75%+3:1, 12 = 75%+M, 13 = 100%+1:1, 14 = 100%+2:1, 15 = 100%+3:1, 16 = 100%+M, PLERR = partial land equivalent ratio of rice, PLERGP = partial land equivalent ratio of rice grass pea.

Although 50% SPGP relay intercropped with rice in a 1:3 SA had the highest combined yield advantage in terms of LER indices (Fig 4), the LER values of all intercropping systems were sufficiently greater than one (Fig 4), indicating that growing the component crops in an intercropping was preferable to growing the component crops separately. According to this conclusion, the solitary cropping method needs 0.83–0.99 more hectares of land in order to provide a yield that is equal to that of the intercropping system, or 83%–99% more yield than sole cropping. In line with this finding, Egbe et al. [56] showed that most intercropping systems were more efficient than solitary cropping at utilising environmental resources for growth and yield creation. Similar outcomes were noted when pea and barley, wheat and field bean, and maize and faba beans were intercropped [57–59], respectively. Regardless of the SA, the cropping systems' TLOY decreased by 4.63% as the SPGP rose from 25 to 100%. As the number of grass pea plants increased, there may have been fierce competition for the soil's remaining moisture, which may have contributed to the yield decline.

The LER is based on the harvested goods rather than the planned yield proportion of the component crops, and it ignores the length of time that the crops are expected to remain in the field [60]. Additionally, it is unclear which crop yield will be used to standardise combination production when LER is estimated [49]. Due to the variability in time required by the component crops of various intercropping systems, ATER allows a more accurate evaluation of the yield advantage of intercropping over sole cropping [54].

ATER values were lower than LER values in all treatments (Table 3 and Fig 4), indicating that LER overestimated resource utilisation compared to ATER. With yield benefits of 33% and 32%, respectively, over solitary cropping, 50% SPGP was repeatedly intercropped with rice in 1:3 (1.33) and 1:2 (1.32) SA, regardless of the values parallel to LER, and greater ATER was attained (Table 3). This might be because intercropping systems, as opposed to solitary crops, can actually offer more effective total resource utilisation and higher overall production [61]. In numerous cereal-legume intercropping systems, including groundnut-cereal fodders [62], barley-pea [55], faba bean-barley [63], bean-wheat [64], lupine-finger millet [54], and finger

**Table 3. Effects of seed proportion of grass pea and spatial arrangement on the production efficiency (ATER, and LER) in the rice-grass pea relay intercropping system.**

| Treatments combination | | Partial LER | | ATER | REY (t ha$^{-1}$) | MAI |
|---|---|---|---|---|---|---|
| SR | - | 1.000 | | 1.000 | 5.00 | 1 |
| SGP | - | | 1.000 | 1.000 | 3.48 | 1 |
| 25% | 1:1 | 0.94 | 1.01 | 1.28 | 4.96 | 1466.27 |
| | 2:1 | 0.92 | 1.05 | 1.31 | 5.17 | 1516.93 |
| | 3:1 | 0.88 | 1.08 | 1.28 | 5.29 | 1498.93 |
| | MRI | 0.89 | 1.04 | 1.28 | 5.12 | 1458.2 |
| 50% | 1:1 | 0.98 | 0.99 | 1.31 | 4.86 | 1375.13 |
| | 2:1 | 0.96 | 1.01 | 1.32 | 4.72 | 1465.6 |
| | 3:1 | 0.94 | 1.05 | 1.33 | 5.14 | 1535.43 |
| | MRI | 0.93 | 1.01 | 1.30 | 4.98 | 1365.4 |
| 75% | 1:1 | 0.90 | 0.94 | 1.22 | 4.60 | 1270.67 |
| | 2:1 | 0.93 | 0.94 | 1.26 | 4.66 | 1342.63 |
| | 3:1 | 0.94 | 0.99 | 1.29 | 4.85 | 1707.53 |
| | MRI | 0.87 | 0.98 | 1.24 | 4.85 | 1333.27 |
| 100% | 1:1 | 0.88 | 0.98 | 1.25 | 4.32 | 1274.4 |
| | 2:1 | 0.93 | 0.93 | 1.24 | 4.56 | 1295.9 |
| | 3:1 | 0.81 | 1.07 | 1.22 | 5.23 | 1395.4 |
| | MI | 0.91 | 0.97 | 1.25 | 4.79 | 1353.03 |

SPGP, seed proportion of grass pea; SA, spatial arrangement; R, rice; GP, grass pea; ATER, area time equivalent ratio; REY, rice equivalent yield; MAI, monitoring advantage index; MRI, mixed relay intercropping; CRGP, competition ratio of grass pea

millet-haricot bean [3], yield improvements have also been observed in comparison to matching solo crops.

When 25% of SPGP was relay intercropped with rice in a 1:3 (5.29 t ha-1) SA, followed by a 1:2 (5.17 t ha-1) SA, the rice equivalent yield for various rice-grass pea relay intercropping systems was higher (Table 3). This resulted from a combination of a higher market price for grass pea and a better yield from the intercropped grass pea component. But in the grass pea-rice relay intercropping system, 50% SPGP planting in 1:3 SA came in top with the highest TLOY (9.89 t ha-1), followed by 50% SPGP planting in 1:2 SA (9.81 t ha-1) (Fig 4). This result supports the findings of Banik et al. [53], who found that intercropping wheat and chickpea produced a greater total land output yield than solely cultivating wheat and chickpea.

**Economic analysis.** All of the intercropping systems in the current study had positive monetary advantage index (MAI) values (Table 3), which showed that these intercropping systems were more profitable than solitary cropping. These findings imply that intercropping could boost system productivity and boost smallholder farmers' income. When 75% of SPGP was relay intercropped with 1:3 SA (1707.53), the financial benefit index for rice-grass pea relay intercropping was higher. These results suggest that when 75% SPGP is planted in 1:3 SA in the research area, it is more economically feasible, suggesting the general viability of grass pea as a relay intercropping with rice. This outcome is consistent with that of Yayeh et al. [54] and Tenaw et al. [65], who found that lupine intercropping with wheat, finger millet, and barley-faba bean yielded higher MAI and greater economic returns than solitary planting. However, by relay intercropping 50% (335,176.79 ETB ha-1) and 25% (330,718.47 ETB ha-1) SPGP with rice in a 1:3 SA, with marginal rate returns (MRR) of 21438% and 9551, respectively, the maximum net benefit was seen. The lowest net benefit (294,659.52 ETB ha$^{-1}$) was recorded when 75% SPGP was relay intercropped with rice in 1:1 SA (Table 4).

**Table 4. Net benefit (NB) and marginal rate of return (MMR) in rice-grass pea relay intercropping as affected by seed proportion of grass pea and spatial arrangement.**

| Treatments | TVC (ETB) | Gross benefit (ETB) | Net benefit (ETB) | MRR (%) | Dominance |
|---|---|---|---|---|---|
| **25%+1:1** | 1191.12 | 320407 | 319215.9 | - | |
| **25% +2:1** | 1278.88 | 328877 | 327598.1 | 9551 | |
| **25%+3:1** | 1377.53 | 332096 | 330718.5 | 3163 | |
| **25%+MI** | 1497.12 | 321830 | 320332.9 | | D |
| **50%+1:1** | 2257.8 | 321207 | 318949.2 | | D |
| **50%+2:1** | 2345.56 | 317660 | 315314.4 | | D |
| **50%+3:1** | 2438.21 | 337615 | 335176.8 | 21438 | |
| **50%+MI** | 2557.8 | 321920 | 319362.2 | | D |
| **75%+1:1** | 3324.48 | 297984 | 294659.5 | | D |
| **Sole rice and grass pea** | 3390.88 | 328312 | 324921.1 | | D |
| **75%+2:1** | 3412.24 | 308611 | 305198.8 | | D |
| **75%+3:1** | 3504.89 | 317462 | 313957.1 | | D |
| **75%+MI** | 3624.48 | 307611 | 303986.5 | | D |
| **100%+1:1** | 4390.88 | 300751 | 296360.1 | | D |
| **100%+2:1** | 4478.64 | 302870 | 298391.4 | | D |
| **100%+3:1** | 4571.29 | 320703 | 316131.7 | | D |
| **100%+MI** | 4690.88 | 311847 | 307156.1 | | D |

Where, TVC = Total variable cost, MRR = Marginal rate of return, ETBB = Ethiopian birr

## Conclusion

The study confirmed that seed proportion of grass pea and spatial arrangement in additive series relay intercropping of grass pea with low-land rice had no significant effect on rice. The highest grain yield of grass pea was obtained when 25% SPGP was relay intercropped with rice in 1:3 SA. The LER and ATER values of all intercropping systems were sufficiently greater than one, indicating that growing the component crops in an intercropping was advantageous than growing the component crops individually. The rice equivalent yield for different rice-grass pea relay intercropping systems was higher when 25% of SPGP was relay intercropped with rice in 1:3. Although, the monetary advantage index was higher in rice-grass pea relay intercropping when 75% of SPGP was relay intercropped with 1:3 SA, the monetary advantage index (MAI) values were positive in all intercropping systems. The highest net benefit was recorded when 50% SPGP was relay intercropped with rice in 1:3 SA. In conclusion, maximum production efficiency in terms of TLOY and land use efficiency, NB, MRR, and positive MAI with lower CR was obtained when 50% SPGP was intercropped with the full seed rate of rice in 1:3 SA, indicating that this cropping system is a far better production system as compared to what farmers currently use (mixed intercropping system). Thus, this mixture seems to contribute to the development of sustainable crop production systems with limited external inputs. The following research gaps were suggested for further research: (i) the experiment needs to be repeated across locations and years as the experiment was conducted in a specific area and in one growing season; (ii) the effect of grass pea on soil fertility needs to be investigated as rice (the main crop) and grass pea (a supplementary crop) relay intercropping is the dominant cropping system in the study area; (iii) rice intercropping with other staple legume crops needs to be tested to intensify the production efficiency and profitability of the cropping system.

## Supporting information

**S1 File. Data set up for normal statistical analysis (grass pea).**
(DOCX)

**S2 File. Data set up for orthogonal analysis (Rice).**
(DOCX)

## Acknowledgments

This research was done in collaboration with the Ethiopian Ministry of Education. We wish to thank all people involved during data collection, writing, and publication of this paper, particularly the individual farmers who rent us their farm land.

## Author Contributions

**Conceptualization:** Endeshew Assefa, Yayeh Bitew.

**Data curation:** Endeshew Assefa, Yayeh Bitew.

**Formal analysis:** Endeshew Assefa, Yayeh Bitew.

**Funding acquisition:** Endeshew Assefa, Yayeh Bitew.

**Investigation:** Endeshew Assefa, Yayeh Bitew.

**Methodology:** Endeshew Assefa, Yayeh Bitew.

**Project administration:** Endeshew Assefa, Yayeh Bitew.

**Resources:** Endeshew Assefa, Yayeh Bitew.

**Software:** Yayeh Bitew.

**Supervision:** Endeshew Assefa, Yayeh Bitew.

**Validation:** Endeshew Assefa, Yayeh Bitew.

**Visualization:** Endeshew Assefa, Yayeh Bitew.

**Writing – original draft:** Endeshew Assefa, Yayeh Bitew.

**Writing – review & editing:** Endeshew Assefa, Yayeh Bitew.

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
