## [Decision Letter · Decision Letter 0]

25 May 2023

PONE-D-23-01912Enhancing additive series relay intercropping of grass pea (Lathyrus sativus L.) with low land rice (Oryza sativa L.) in North-Western Ethiopia: A farmer’s indigenous knowledgePLOS ONE

Dear Dr. Bitew,

Thank you for submitting your manuscript to PLOS ONE. After careful consideration, we feel that it has merit but does not fully meet PLOS ONE’s publication criteria as it currently stands. Therefore, we invite you to submit a revised version of the manuscript that addresses the points raised during the review process.

We look forward to receiving your revised manuscript.

Kind regards,

Rupesh Kailasrao Deshmukh, Ph.D.

Academic Editor

PLOS ONE

Journal Requirements:

https://www.tandfonline.com/doi/abs/10.1080/00103624.2022.2043349?journalCode=lcss20

https://file.scirp.org/Html/19-2601113_44714.htm

https://www.researchgate.net/publication/258338423_Yield_and_Competition_Indices_of_Intercropping_Cotton_Gossypium_hirsutum_L_Using_Different_Planting_Patterns

In your revision ensure you cite all your sources (including your own works), and quote or rephrase any duplicated text outside the methods section. Further consideration is dependent on these concerns being addressed.

Reviewers' comments:

Reviewer's Responses to Questions

**Comments to the Author**

1. Is the manuscript technically sound, and do the data support the conclusions?

Reviewer #1: Partly

Reviewer #2: Yes

2. Has the statistical analysis been performed appropriately and rigorously? 

Reviewer #1: Yes

Reviewer #2: Yes

3. Have the authors made all data underlying the findings in their manuscript fully available?

Reviewer #1: Yes

Reviewer #2: Yes

4. Is the manuscript presented in an intelligible fashion and written in standard English?

Reviewer #1: No

Reviewer #2: No

5. Review Comments to the Author

Reviewer #1: he entitled manuscript “Enhancing additive series relay intercropping of grass pea (Lathyrus sativus L.) with low land rice (Oryza sativa L.) in North-Western Ethiopia: A farmer’s indigenous knowledge”.

The submitted manuscript emphasised that seed proportion of grass pea and spatial arrangement had no significant effect on rice crop in additive series relay intercropping of grass pea with low land rice.

However, the present study trial has also been conducted in one season in single location in Ethiopia, Therefore, the reliability of the results is doubtful and cannot make conclusion on the basis of it. Therefore, it is suggested that at least Two year and multi-location data should be included in the present submitted manuscript. Apart from this, the present manuscript need to rectify the grammatical as well as typological mistakes.

Observations:

1. Please check the sentence “Banik et al. [53], who observed that the chick pea yield in wheat-check pea mixture significantly lower than sole cropped check pea.”

2. Please correct the words Results “Results revealed that the CR of rice was greater”

3. Too lengthy sentences and not clear “Results revealed that partial land equivalent ratio (PLER) of rice and grass pea in all cropping systems were greater than 0.5 (Figure 4 ) indicating that there was null or very low growth resource competition between the component crops during their co-growing period which supports the CR values.

Reviewer #2: The article needs to be modified according to the comments ( Please find the attachment )

The introduction and discussion part need to be improvised with the adsition of related articles and findings.

6. PLOS authors have the option to publish the peer review history of their article (what does this mean?). If published, this will include your full peer review and any attached files.

Reviewer #1: No

Reviewer #2: No

---

## [Author Response · Author response to Decision Letter 0]

30 May 2023

1) Grammetical errors need to be corrected.

Response: fully checked ad corrected y grammar checker

2) Introduction part needs to be improvised accrording to the title of the research paper.

Response: Updated Based on the comment

3) The different planting methods and patterns can be displayed in a table form in the materials and methods.

Response: Clearly descried IN the document

Table 2. Treatment combination used in additive series relay intercropping of grass pea with rice

Trt No. SPGP (%) SA (R: GP) TCS

1 25 1:1 Row intercropping

2 25 2:1 Row intercropping

3 25 3:1 Row intercropping

4 25 MRI Mixed intercropping

5 50 1:1 Row intercropping

6 50 2:1 Row intercropping

7 50 3:1 Row intercropping

8 50 MRI Mixed intercropping

9 75 1:1 Row intercropping

10 75 2:1 Row intercropping

11 75 3:1 Row intercropping

12 75 MRI Mixed intercropping

13 100 1:1 Row intercropping

14 100 2:1 Row intercropping

15 100 3:1 Row intercropping

16 100 MRI Mixed intercropping

17 Sole rice - Row planting

18 Sole grass pea - Broadcasting

Trt No., treatment number; SPGP (%), seed proportion of grass pea; SA, spatial arrangement; TCS, type of cropping system; GP, Grass pea; R, rice; R: GP, rice to grass pea seed proportions; MRI, mixed relay intercropping.

4) Results are highlighted only for some parameters. May explained in detail.

Response: Yes, because the main objectives of this work is to evaluate the lad use efficiency usig different competition indices

5) Discussion part need to be modified more precisely.

6) Conclusion doesn't highlight the whole research. It need to be revised.

Response: The study confirmed that seed proportion of grass pea and spatial arrangement in additive series relay intercropping of grass pea with low-land rice had no significant effect on rice. The highest grain yield of grass pea was obtained when 25% SPGP was relay intercropped with rice in 1:3 SA. The LER and ATER values of all intercropping systems were sufficiently greater than one, indicating that growing the component crops in an intercropping was advantageous than growing the component crops individually. The rice equivalent yield for different rice-grass pea relay intercropping systems was higher when 25% of SPGP was relay intercropped with rice in 1:3. Although, the monetary advantage index was higher in rice-grass pea relay intercropping when 75% of SPGP was relay intercropped with 1:3 SA, the monetary advantage index (MAI) values were positive in all intercropping systems. The highest net benefit was recorded when 50% SPGP was relay intercropped with rice in 1:3 SA. In conclusion, maximum production efficiency in terms of TLOY and land use efficiency, NB, MRR, and positive MAI with lower CR was obtained when 50% SPGP was intercropped with the full seed rate of rice in 1:3 SA, indicating that this cropping system is a far better production system as compared to what farmers currently use (mixed intercropping system). Thus, this mixture seems to contribute to the development of sustainable crop production systems with limited external inputs. The following research gaps were suggested for further research: (i) the experiment needs to be repeated across locations and years as the experiment was conducted in a specific area and in one growing season; (ii) the effect of grass pea on soil fertility needs to be investigated as rice (the main crop) and grass pea (a supplementary crop) relay intercropping is the dominant cropping system in the study area; (iii) rice intercropping with other staple legume crops needs to be tested to intensify the production efficiency and profitability of the cropping system.

Comments to the Author

1. Is the manuscript technically sound, and do the data support the conclusions?

Reviewer #1: Partly

Reviewer #2: Yes

Response:

2. Has the statistical analysis been performed appropriately and rigorously? 

Reviewer #1: Yes

Reviewer #2: Yes

3. Have the authors made all data underlying the findings in their manuscript fully available?

Reviewer #1: Yes

Reviewer #2: Yes

4. Is the manuscript presented in an intelligible fashion and written in standard English?

Reviewer #1: No

Reviewer #2: No

Response: updated

5. Review Comments to the Author

Reviewer #1: he entitled manuscript “Enhancing additive series relay intercropping of grass pea (Lathyrus sativus L.) with low land rice (Oryza sativa L.) in North-Western Ethiopia: A farmer’s indigenous knowledge”.

The submitted manuscript emphasised that seed proportion of grass pea and spatial arrangement had no significant effect on rice crop in additive series relay intercropping of grass pea with low land rice.

However, the present study trial has also been conducted in one season in single location in Ethiopia, Therefore, the reliability of the results is doubtful and cannot make conclusion on the basis of it. Therefore, it is suggested that at least Two year and multi-location data should be included in the present submitted manuscript. Apart from this, the present manuscript need to rectify the grammatical as well as typological mistakes.

Response: further research is suggested ad the manuscript was updated grammatical

Observations:

1. Please check the sentence “Banik et al. [53], who observed that the chick pea yield in wheat-check pea mixture significantly lower than sole cropped check pea.”

Response: This result contradicted the results of Banik et al. [53], who observed that the chickpea yield in a wheat-chickpea mixture was significantly lower than that of a solely cropped chickpea.

2. Please correct the words Results “Results revealed that the CR of rice was greater”

3. Too lengthy sentences and not clear “Results revealed that partial land equivalent ratio (PLER) of rice and grass pea in all cropping systems were greater than 0.5 (Figure 4 ) indicating that there was null or very low growth resource competition between the component crops during their co-growing period which supports the CR values.

Response: corrected based on the comment

Reviewer #2: The article needs to be modified according to the comments ( Please find the attachment )

The introduction and discussion part need to be improvised with the adsition of related articles and findings.

Response: corrected based on the comment

---

## [Decision Letter · Decision Letter 1]

22 Jun 2023

Enhancing the land use efficiency of low-land rice ( Oryza sativa L.) - grass pea (Lathyrus sativus L.) additive series relay intercropping in North-Western Ethiopia: A farmer’s indigenous knowledge

PONE-D-23-01912R1

Dear Dr. Bitew,

We’re pleased to inform you that your manuscript has been judged scientifically suitable for publication and will be formally accepted for publication once it meets all outstanding technical requirements.

Kind regards,

Rupesh Kailasrao Deshmukh, Ph.D.

Academic Editor

PLOS ONE

Additional Editor Comments (optional):

Reviewers' comments:

Reviewer's Responses to Questions

**Comments to the Author**

1. If the authors have adequately addressed your comments raised in a previous round of review and you feel that this manuscript is now acceptable for publication, you may indicate that here to bypass the “Comments to the Author” section, enter your conflict of interest statement in the “Confidential to Editor” section, and submit your "Accept" recommendation.

Reviewer #1: All comments have been addressed

Reviewer #2: All comments have been addressed

2. Is the manuscript technically sound, and do the data support the conclusions?

Reviewer #1: Yes

Reviewer #2: Yes

3. Has the statistical analysis been performed appropriately and rigorously? 

Reviewer #1: Yes

Reviewer #2: Yes

4. Have the authors made all data underlying the findings in their manuscript fully available?

Reviewer #1: Yes

Reviewer #2: Yes

5. Is the manuscript presented in an intelligible fashion and written in standard English?

Reviewer #1: Yes

Reviewer #2: Yes

6. Review Comments to the Author

Reviewer #1: (No Response)

Reviewer #2: (No Response)

7. PLOS authors have the option to publish the peer review history of their article (what does this mean?). If published, this will include your full peer review and any attached files.

Reviewer #1: **Yes: **Surendra Barpete

Reviewer #2: **Yes: **Dharini Chittaragi

---

## [Editor Report · Acceptance letter]

27 Jun 2023

PONE-D-23-01912R1 

Enhancing the land use efficiency of low-land rice *(Oryza sativa L.)* - grass pea *(Lathyrus sativus L.)* additive series relay intercropping in North-Western Ethiopia: A farmer’s indigenous knowledge 

Dear Dr. Bitew:

I'm pleased to inform you that your manuscript has been deemed suitable for publication in PLOS ONE. Congratulations! Your manuscript is now with our production department. 

Kind regards, 

on behalf of

Dr. Rupesh Kailasrao Deshmukh 

Academic Editor

PLOS ONE